# A black carbon peak and its sources in the free troposphere of Beijing induced by cyclone lifting and transport from Central China

Zhenbin Wang[1234], Bin Zhu[1234*], Hanqing Kang[1234], Wen Lu[1234], Shuqi Yan[1234], Delong Zhao[5], Weihang Zhang[6], Jinhui Gao[7]

[1]Collaborative Innovation Center on Forecast and Evaluation of Meteorological Disaster, Nanjing University of Information Science and Technology, Nanjing, 210044, China
[2]Key Laboratory for Aerosol-Cloud-Precipitation of China Meteorological Administration, Nanjing University of Information Science and Technology, Nanjing, 210044, China
[3]Key Laboratory of Meteorological Disaster, Ministry of Education (KLME), Nanjing University of Information Science and Technology, Nanjing, 210044, China
[4]Joint International Research Laboratory of Climate and Environment Change (ILCEC), Nanjing University of Information Science and Technology, Nanjing, 210044, China
[5]Beijing Weather Modification Office, Beijing, 100089, China
[6]College of Oceanic and Atmospheric Sciences, Ocean University of China, Qingdao, 266100, China
[7]Plateau Atmosphere and Environment Key Laboratory of Sichuan Province, School of Atmospheric Sciences, Chengdu University of Information Technology, Chengdu, 610225, China

*Correspondence to*: B. Zhu (binzhu@nuist.edu.cn)

**Abstract.** Observations suggest that the vertical distributions of air pollutants, such as black carbon (BC), present as various types depending on the emission sources and meteorological diffusion conditions. However, the formation process and source appointment of some special BC profiles are not fully understood. In this paper, by using the Weather Research and Forecasting model coupled with chemistry (WRF-Chem) with a BC-tagging technique, we investigate the formation mechanism and regional sources of a BC peak in the free troposphere observed by aircraft flight in Beijing (BJ) on May 5th, 2018. The results show that the contribution rate of the Beijing-Tianjin-Hebei (BTH) region to the surface BC of BJ exceeded 80% in this case. Local sources dominated BC in BJ from the surface to approximately 700 m (78.5%), while the BC peak in the free troposphere (~4000 m) was almost entirely imported from external sources (99.8%). Combining BC tracking and process analysis, we find that horizontal advection (HADV) and vertical advection (VADV) processes played an important role in the convergent and upward movement and the transport of BC. The BC originating from the surface in central provinces, including Shanxi (SX), Henan (HN) and Hebei (HB), was uplifted through a cyclone system 16 hours previously, transported to a height of approximately 3000 m above BJ, and then lifted by the VADV process to approximately 4000 m. At the surface, BJ and its surroundings were under the control of a weak pressure gradient, leading to the accumulation of BC within the boundary layer. Our results indicate that cyclone systems can quickly lift air pollutants, such as BC, up to the free troposphere, as well as extend their lifetimes and further affect the regional atmospheric environment and climate.

# 1 Introduction

Black carbon aerosol (BC) has been a research hotspot in recent years owing to its significant environmental and climate effects (IPCC, 2013). Unlike scattering aerosols (sulfate, etc.), BC has a strong ability to absorb solar radiation, which accelerates global warming (Jacobson, 2001; Weingartner et al., 2003; Bond et al., 2013). Additionally, BC can heat the upper air of the planetary boundary layer (PBL), inhibit its development, and promote regional air pollution (Ding et al., 2016). Air quality is affected not only by local emissions but also by long-distance transport (Zhang et al., 2017a). For

example, BC in South China is mainly transported from Southeast Asia and North China (Fang et al., 2020), and most of the BC in the Arctic is transported from low-latitude regions (Keegan et al., 2014). Therefore, it is necessary to study the temporal and spatial distributions of BC and quantitatively analyse its regional sources to provide a reference for BC emission reduction.

The annual carbon emissions in China account for approximately one-quarter of the global carbon emissions (Bond et al.,

2004). As the capital of China, Beijing (BJ) has experienced frequent air pollution in recent years (Van Pinxteren et al., 2009). In BJ and its surrounding areas, straw is often burned in early summer; hence, BC pollution has become an environmental problem that cannot be ignored (Bergin et al., 2001; He et al., 2001). However, most of the previous measurements of BC were performed at ground level, and the quantitative source tracking of BC was not thorough enough. For example, based on radiocarbon ($^{14}$C) measurements, Zhang et al. (2015) found that fossil emissions dominated BC with a

mean contribution of 75±8% in BJ, but regional source research was lacking. To identify the regional sources, Liu et al. (2018) used a potential source contribution function (PSCF) model to determine that SX and HB were the main sources of BC in BJ but failed to quantify their separate contributions. Wen et al. (2020) used the Weather Research and Forecasting model coupled to chemistry (WRF-Chem) to simulate the regional contribution and found that regional transport contributes more than 40% to BC in BJ, but there was no detailed delineation of external source regions.

In addition to research on horizontal BC, many vertical observations have also been conducted in recent years, showing that vertical distributions of BC are strongly affected by meteorological conditions and regional transport. The aircraft observation experiment carried out in BJ showed that regional transport of BC can enhance air pollution, and transport occurs not only near the surface but also in the middle levels of the PBL (0.5-1 km) (Zhao et al., 2015). Moreover, when the BC profile has a homogeneous or negative gradient distribution, the diurnal evolution of the PBL is the leading factor. When

there is a positive BC gradient from the surface to the top of the PBL, it is mainly caused by surrounding emissions from high stacks and regional transport (Lu et al., 2019; Shi et al., 2021). However, these studies were mostly qualitative inferences, and it is difficult to quantitate the formation process of vertical distributions and BC sources in detail. In addition, the effect of BC on PBL varies greatly owing to its altitude. BC will inhibit the development of the PBL above the morning residual layer and conversely promote its development (Ma et al., 2020). Therefore, it is worth emphasizing that using only

BC surface concentrations to calculate radiative forcing and heating rate and ignoring the vertical distribution of BC will induce great uncertainties (Shi et al., 2021).

In summary, there are few studies on the quantitative interpretation of the regional contribution of BC, or the source regions are not sufficiently separated; additionally, research on the reasons and source tracking of the vertical distribution of BC is even scarcer. Based on the BC profile observed in BJ, this study uses the air quality model WRF-Chem with a BC-tagging technique to track the sources of BC, with the hope to explain the observed BC peak in the free troposphere and evaluate BC sources and direct BC regional emission reduction measures.

## 2 Data and methodology

### 2.1 Description of data used

### 2.1.1 Data used for modelling

A variety of data were used for modelling in this study. The initial and boundary meteorological and chemical conditions were provided by the National Centers for Environmental Prediction (NECP) final (FNL) operational global analysis data and outputs of the Community Atmosphere Model with chemistry (CAM-chem; Lamarque et al., 2012). Anthropogenic emissions were provided by the Multiresolution Emission Inventory for China (MEIC, 2016, http://www.meicmodel.org/). Biogenic emissions were generated by using the Model of Emissions of Gas and Aerosols from Nature (MEGAN; Guenther et al., 2006).

### 2.1.2 Ground-based observation

In terms of the observation data used for model verification, the meteorological factors were derived from ground station observations using the Meteorological Information Comprehensive Analysis and Processing System (MICAPS) with a time accuracy of 3 h. $PM_{2.5}$ data were obtained from the China air quality online monitoring and analysis platform (https://www.aqistudy.cn/) provided by the Ministry of Ecology and Environment of China (http://106.37.208.233:20035/), which includes 1500 monitoring stations. Each site was equipped with tapered element oscillating microbalance instruments (TEOM, RP1400 Model, Thermo Scientific Company, USA) using the micro-oscillating balance method and the β-absorption method to measure $PM_{2.5}$ concentration at a resolution of 0.1 $\mu g \cdot m^{-3}$ (Zhang and Cao, 2015). More details about the measurement methods, accuracy, and uncertainties are described in China Environmental Protection Standards HJ 653-2013 (http://www.cnemc.cn/jcgf/dqhj/201711/W020181008687887167307.pdf). Many studies on air pollution, particularly $PM_{2.5}$ pollution, have obtained their $PM_{2.5}$ data from this platform, such as Kang et al. (2021) and Hou et al. (2019, 2020). BC data were collected by the single particle soot photometer (SP2, Droplet Measurement Technologies, Inc., USA) that was equipped with an external inlet, the Model 1200 passive isokinetic inlet (Brechtel Manufacturing Inc., USA), which could deliver a sample flow of 150 lpm with an air speed of 100 $m \cdot s^{-1}$ with a sample efficiency of 95% for particle sizes between 0.01 and 6 $\mu m$. More details about SP2 can be found in Sharma et al. (2017) and Stephens et al. (2003).

### 2.1.3 Aircraft observation platform

In May 2018, we carried out a total of 7 aircraft observations in BJ and surrounding areas (Figure S1), with each observation time between 10:00 and 12:00 (Beijing Time, BJT). The airborne measurements in this study were performed on a King Air 350 aircraft observation platform with a true speed of approximately 250-300 km/h. An Aircraft Integrated Meteorological Measurement System (AIMMS-20, Aventech, Canada) was used to measure meteorological parameters in situ, including temperature (T), relative humidity (RH), wind speed (WS) and wind direction (WD), with a time resolution of 1s. An SP2 was also installed on the aircraft observation platform to measure the vertical distribution of BC. The operation of most flights was carried out to avoid clouds where possible, which could effectively avoid the wet deposition of BC. More detailed information about the aircraft platform can be found in Tian et al. (2019), Hu et al. (2020) and Zhao et al. (2019&2020).

### 2.2 Model description

The model used in this study, WRF-Chem 3.9.1.1, is coupled with a BC-tagging technique (Wang et al., 2014; Yang et al., 2017; Yang et al., 2018; Fang et al., 2020; Zhu et al., 2020), which is similar to the particle source analysis technology in the Comprehensive Air Quality Model with Extensions (CAMx) model. The model can track the BC emitted from the predivided source region and quantitatively calculate the BC concentration in each source region one time. We divide the model domain into 20 geographic source regions before running the model, so the BC concentration of the target region is equal to the sum of the concentrations of BC from all source regions:

$$C = \sum_{i=1}^{20} C_i \quad (1).$$

The model domain and the administrative regions of each source region are shown in Fig. 1 and Table 1, respectively (the source regions with less contribution will be merged in the subsequent analysis, defined as Others). The BC concentration in each source region can be expressed as follows:

$$\Delta C_i = \Delta Chem_i + \Delta Phy_i + \Delta Emis_i \quad (2),$$

where $\Delta Chem_i$, $\Delta Phy_i$ and $\Delta Chem_i$ represent the BC concentration produced by chemical, physical and emission processes, respectively. However, BC is not included in the calculation of chemical reactions, so the value of $\Delta Chem_i$ is 0. $\Delta Phy_i$ represents the concentration of BC changed by model physical processes, including horizontal advection, vertical advection, vertical mixing, dry deposition, wet deposition and convection. The BC of each source region is regarded as an independent variable, and it is marked from the beginning of the BC emission, $\Delta Emis_i$. The concentration of the total emissions in region i is defined as follows:

$$\Delta Emis_i(x, y, z) = \begin{cases} \Delta Emis_i(x, y, z) & \text{inside region } i \\ 0 & \text{outside region } i \end{cases} \quad (3).$$

If BC is in region i, then $\Delta Emis_i (x, y, z) = \Delta Emis (x, y, z)$; otherwise, $\Delta Emis_i (x, y, z) = 0$. Subsequently, the newly defined variables will participate in the physical and chemical calculation process in the model accompanied by the origin variables in the model. Thus, we can obtain the BC concentrations of each source region at any grid of the model and at any time.

Compared with sensitive experiments, this method can more accurately quantify the sources of BC with fewer errors. Previous studies have used a similar technique to study the source of air pollutants, such as $PM_{2.5}$ and $O_3$, and the results all show that regional transport is an important factor of $PM_{2.5}$ and $O_3$ pollution in BJ (Gao et al., 2016; Zhang et al., 2017b; Gao et al., 2020). In addition, the air pollutants in BJ are mainly from BJ, TJ and HB, and the high concentration area presents a banded distribution feature from southwest to northeast, which is consistent with the analysis of BC in this study.

## 2.3 Parameterized scheme settings

Experiments started at 08:00 BJT on 01 May 2018 and ended at 08:00 BJT on 10 May 2018. The first 3 days were designated the spin-up time. Regarding the simulation settings, two nested domains (Fig. 1) were set up with grid sizes of 99 × 99 at horizontal resolutions of 36 and 12 km for D01 and D02, respectively. D01 covered most parts of China, the surrounding areas and ocean, and D02 covered most parts of North China. The modelling results of D01 provided

meteorological and chemical boundary conditions for the simulations of D02. For the vertical direction, 38 layers were set up from the surface up to a pressure limit at 50 hPa, where 10 layers were located below 1 km. Notably, the carbon bond mechanism Z (CBM-Z, Zaveri and Peters, 1999) was applied as the gas-phase chemical mechanism in this study. Correspondingly, the Model for Simulating Aerosol Interactions and Chemistry with eight bins (MOSAIC-8bins; Zaveri et al., 2008) was chosen as the aerosol chemistry mechanism. Other parameterization settings are listed in Table 2.

## 3 Results and discussion

### 3.1 Model validation

Although the WRF-Chem model is widely used in air quality research, there are significant differences in the simulation results with different parameterization schemes. In this study, time series of both simulated and observed $PM_{2.5}$, BC concentration, temperature (T) and wind (Ua and Va) are shown in Fig. 2 to evaluate the performance of the model. Figure 2

illustrates that the model reproduces the numerical magnitude and variation characteristics of $PM_{2.5}$, BC and meteorological factors well. Additionally, we calculate several common metrics (correlation coefficient (R), index of agreement (IOA), mean bias (MB), root mean square error (RMSE), mean normalized bias (MNB), mean fractional bias (MFB) and total error (TE)) to validate the model performance on meteorological factors and air pollutants. The benchmarks shown in brackets in Table 3 follow the recommended values suggested by Emery and Tai (2001) and EPA (2007). Generally, the model

reproduces the pollutants and main meteorological elements well, which provides a good basis for subsequent analysis. The model validation in North China, particularly in HN and SX (source regions), is presented in Figures S2 and S3 and Tables S1 and S2 in the supplement, and the modelling profiles of BC and meteorological factors are also presented in Fig. 6 in Section 3.2.2.

As shown in Table 3, meteorological factors (T, Ua and Va) showed high values of the mean IOAs ($\geq 0.75$), indicating that the simulation agreed well with the observations. The MB of T was -0.25, which suggested that the mean bias of T was within 1 ℃, and the MB of Ua and Va were -0.36 and -0.01, respectively, suggesting that the model results deviated from the observations to a small extent. The RMSE and TE of meteorological elements were less than 1.88 and 1.44 respectively, which are comparable with the values reported in another modelling study (Gao et al., 2020) and were both within the threshold range. However, the MNB and MFB of Ua were beyond the scope of its benchmark, suggesting an overestimation of Ua. The MNB and MFB of the other meteorological factors were within the scope of their benchmark.

For air pollutants, good agreement was found between the simulations and observations since the IOAs of $PM_{2.5}$ and BC were 0.93 and 0.70, respectively. The MB values of BC and $PM_{2.5}$ were -0.42 and -11.2, respectively, which demonstrated that the deviation in MB between BC models and observations was less than 1 $\mu g \cdot m^{-3}$, and the deviation in $PM_{2.5}$ was approximately 11 $\mu g \cdot m^{-3}$. Notably, all the metrics of air pollutants also indicated that the model performance for air pollutants was acceptable.

## 3.2 Tracking BC from the surface to the free troposphere

During the aircraft observation on May 5th, 2018, we found high BC values both near the ground and in the free troposphere (Fig. 3). To reveal the cause of the distinct BC profile, the WRF-Chem model with a BC-tagging technique was used for quantitative explanation.

### 3.2.1 Tracking BC sources at the surface

The near-surface pollution in BJ lasted from May 5th to 6th (Fig. 2), and both the spatial and temporal distributions (near-surface and in the free troposphere) of BC and $PM_{2.5}$ exhibited highly similar characteristics (Fig. S4). In the synoptic chart of Fig. 4a, there was a weak cyclone system in SX and NWCHN at 17:00 BJT on May 4th, leading to convergence and upward movement there. At 08:00 BJT on May 5th (Fig. 4b), the regions controlled by the weak cyclone system had expanded, including SX, NWCHN, HB and HN, while BJ was in front of the weak low pressure. At the same time, it can be seen that the near-surface convergence areas were mainly in SX from May 4th to 5th (Fig. 4d and e), corresponding to the position where the cyclone appeared, indicating that convergence uplift of surface BC existed in SX, which was consistent with the cross-section analysis in Section 3.3.2. There was convergence near the surface, and air pollutants in the surrounding areas were likely to accumulate, leading to an increase in BC concentration in BJ (Fig. 4d and e). Moreover, there was almost no convergence but an obvious 'transport channel' from SX to BJ in the free troposphere (Fig. 4h), so BC was transported from SX to BJ by westerly winds. Subsequently, the low-pressure system moved south, and the BJ was controlled by the uniform pressure field at 08:00 BJT on May 6th (Fig. 4c), which was not beneficial for the diffusion of BC in the horizontal and vertical directions (Fig. 4f). At an altitude of approximately 4000 m, BC was transported southwards due to clean north winds (Fig. 4i).

Moreover, the sources of BC were quantitatively traced (Fig. 5) to explain the cause of pollution in BJ. The mean BC concentration was 2.29 μg·m$^{-3}$ in this case (Fig. 5a), and the diurnal evolution of BC showed a characteristic of 'lower in the daytime and higher in the nighttime', which is related to the daily change in the PBL (Huang et al., 2018). Furthermore, BC near the surface mainly originated from BJ, HB, HN, TJ and SD, whose mean contribution rates were 52.0%, 23.9%, 6.1%, 5.7% and 4.4%, respectively (Fig. 5b). The BC contribution rate of the BTH region to BJ exceeded 80%, further confirming that when controlled by weak low pressure, air pollutants in surrounding areas are likely to accumulate in BJ, and local pollutants have difficulty diffusing outward (Chen et al., 2008).

### 3.2.2 Tracking BC sources vertically

Near-surface BC mainly originated from BJ and its surrounding areas, while the source of BC in the free troposphere (~4000 m) showed different characteristics. The vertical distributions of BC and meteorological factors were well simulated, especially BC, T and RH (Fig. 6a and b). However, the WS under 2 km and WD at 1.25 km did not well show agreements between SIM and OBS, which could be attributed to local circulations such as land-sea and valley breezes across the complex terrain (Igel et al., 2018; Quan et al., 2020; Zhang et al., 2021). In this study, we focus more on the effect of upper-level winds on the transport of BC, and the winds at the altitude of 4000 m are significantly better modelled than the winds within the boundary layer. Overall, the model results are also acceptable in the vertical direction. The BC concentration presents a decreasing trend from the ground to approximately 700 m but increases from approximately 3000 m, forming a peak at an altitude of approximately 4000 m (Fig. 6a). As shown in Fig. 6e, the total contribution rate of BJ and HB below 700 m was as high as 96.7%, which is related to the easterly wind near the surface (Fig. 6d). For the BC peak in the free troposphere (~4000 m), almost all BC was transported from external regions, including SX, NWCHN, SWCHN, HB, HN, and even SCHN, and the contribution rates were 24.7%, 23.5%, 10.5%, 9.4%, 8.9 and 7.1%, respectively. In addition, Fig. 6d illustrates that this altitude was mainly controlled by westerly winds (~225°), and the wind speeds increased significantly above 3000 m, which is beneficial to the long-distance transport of BC.

### 3.3 Air mass trajectory and physical process analysis

### 3.3.1 Air mass trajectory analysis

To determine the source of BC in depth and to validate the source tracing results above, we utilized the Hybrid Single Particle Lagrangian Integrated Trajectory (HYSPLIT) model, developed by the National Oceanic and Atmospheric Administration (NOAA), to analyse the backward trajectory (24 h) of air masses reaching the altitudes of 600, 2200 and 3800 m above BJ, with intervals of 1600 m, at 08:00 BJT on May 5th (Fig. 7). As presented in Fig. 7, the air mass near the surface (red line) originated from BJ and its surrounding areas, which is consistent with the source tracking results (Fig. 6e). The upper air mass (green line) originated in SX and reached BJ via HB, corresponding to the results of the cross section (Fig. 6e, Fig. 8). In addition, the air mass reaching at the altitude of approximately 2000 m above BJ (blue line) was from

Mongolia, a source region outside of D02, so the BC source tracing results in a predominantly Other source region (Fig. 6e, the Other source region includes Mongolia and Inner Mongolia). Furthermore, the green line indicates that from 08:00 BJT on May 4th to 08:00 BJT on May 5th, there was an air mass lifting from the surface in SX along a southwestern path and reaching the upper levels of BJ (~4000 m). During this time, the conditions in Central China (including SX, HB and HN) were controlled by a cyclone system (Fig. 4a and b), leading to convergent and elevating motion there, which is conducive to the uplift of near-surface air masses and consistent with the backward trajectory model results.

### 3.3.2 Tracking BC sources in the cross section

In addition to the backward trajectory, the BC sources in the vertical cross section along the path from SX to BJ (in Figs. 1 and 7) can more convincingly explain the BC uplift and transport phenomenon. Referring to the backward trajectory, the white line shown in Fig. 1b can be regarded as the transport path of the air mass. To illustrate the uplift and transport mechanism of BC, cross sections of BC concentration, wind vector, and BC contour lines (including BJ, SX and NWCHN) along the aforementioned line are shown in Fig. 8. From 08:00 to 18:00 BJT on May 4th, the southwest wind prevailed along the SX-BJ line (Fig. 8a and b), which contributed to the transport of BC. BC originated in the NWCHN transported to SX and mixed with BC emitted in the SX region. By 16:00 BJT on May 4th, the ascending movement in SX was vigorous, resulting in the elevation of near-surface BC (Fig. 8b), which is consistent with the initial uplifting position presented in Fig. 7. BC originating from NWCHN and SX was uplifted up to 3000 m and transported to HB and BJ to form high values (Fig. 8c). However, the wind direction near the surface changed from westerly to easterly, while westerly winds still prevailed in the upper air (Fig. 8d). As a result, near-surface BC in BJ diffused to the surrounding areas, such as HB, but in the upper air, BC originating in NWCHN and SX was still transported to BJ and lifted further to approximately 4000 m (Fig. 8d). Thus far, there were high BC values both near the surface and in the free troposphere above BJ (~4000 m), which is consistent with the observation and source tracing results in Fig. 6.

### 3.3.3 Physical process analysis

The backward trajectory shows that the air mass in the free troposphere originated from SX, while that within the boundary layer was from BJ; therefore, we further quantified the dominant transport processes of BC in the source region (SX) and receptor region (BJ), including horizontal advection (HADV), vertical advection (VADV) and vertical mixing (VMIX). In Fig. 9, the results show that HADV and VADV played an important role in the convergent and upward movement induced by cyclones and the transport of BC. From 14:00 to approximately 18:00 BJT on May 4th, in the boundary layer (<2 km), HADV made positive contributions to BC concentration in SX, while the VADV had the opposite effect, suggesting that convergent and upward motion existed therein (Fig. 9a and b), which also corroborates the cyclone system in the surface synoptic patterns and near-surface convergence areas (Fig. 4a and d). At the same time, in the free troposphere (~4 km), the contribution of VADV and HADV to BC concentration was opposite to that in the boundary layer, indicating that BC originating in SX was lifted from the surface to the free troposphere by VADV (Fig. 9a and b), which is consistent with the

backward trajectory and cross-sectional analysis (Fig. 8b and c). In the upper layer of BJ (~3 km), HADV made a positive

contribution to the BC concentration from 14:00 to approximately 22:00 BJT on May 4[th], showing that BC was transported from SX by westerly winds, which is also consistent with the analysis of BC transport in Figure 4h. Then, VADV lifted the BC transported by the HADV to approximately 4 km from 02:00 to approximately 08:00 BJT on May 5[th] (Fig. 9d and e), consistent with the analysis in Fig. 8c and d. VMIX in both SX and BJ occurred mainly in the afternoon of May 4[th] owing to intense turbulence (Fig. 9c and f).


## 4 Conclusions

This paper utilizes the air quality model WRF-Chem with a BC-tagging technique to study the formation mechanism of a special BC profile observed by an aircraft flight in BJ. The major findings are summarized as follows:

In this case, the mean BC concentration was 2.29 $\mu g \cdot m^{-3}$. BC at the surface mainly originated from BJ, HB, HN, TJ and SD,

with contribution rates of 52.0%, 23.9%, 6.1%, 5.7% and 4.4%, respectively. The BC contribution rate of the BTH region to BJ exceeded 80%, further confirming that when controlled by weak low pressure, air pollutants in surrounding areas are likely to accumulate in BJ, and local air pollutants have difficulty diffusing outward.

Local sources dominated BC in BJ from the surface to approximately 700 m (78.5%), while BC was almost entirely imported from external sources (99.8%) in the free troposphere (~4000 m). BC in the free troposphere mainly originated

from SX, HB, HN, and NWCHN, and the contribution rates were 24.7%, 9.4%, 8.9% and 23.5%, respectively.

Figure 10 illustrates the formation mechanism of the special BC profile. HADV and VADV processes played an important role in the convergent and upward movement and the transport of BC. Near-surface BC that originated from SX, HB, HN and NWCHN was uplifted by a cyclone system approximately 16 hours previously, transported to a height of approximately 3000 m above BJ, and then lifted by the VADV process to approximately 4000 m. At the surface, BJ and its surroundings

were in the field of a weak pressure gradient, leading to the accumulation of BC.

The results indicate that cyclone systems can quickly lift air pollutants, such as BC, up to the free troposphere, extend their lifetimes, and further affect the regional atmospheric environment and climate.

*Code and data availability.* All the observations and model outputs mentioned in this study are publicly available. Observations of $PM_{2.5}$ concentrations can be downloaded directly via the real-time release platform of the Ministry of
Ecology and Environment of China (http://www.mee.gov.cn/). Other observations and the simulated results can be accessed by contacting Zhenbin Wang at wangzb@nuist.edu.cn.

*Author contributions.* ZW performed the model simulation, data analysis and manuscript writing. BZ proposed the idea, supervised this work and revised the manuscript. DZ provided the observation data at BJ station. WL provided help with the model simulation. HK, SY and WZ also contributed to the manuscript revision.

*Competing interests.* The authors declare that they have no conflict of interest.

*Acknowledgements.* We acknowledge the free use of MEIC emissions from Tsinghua University (http://www.meicmodel.org/dataset-mix.html). We are grateful to the High Performance Computing Center of Nanjing University of Information Science and Technology for performing the numerical calculations in this work on its blade cluster system.

*Financial support.* This work was supported by the National Key Research and Development Program (grant no. 2016YFA0602003) and National Natural Science Foundation of China (grant nos. 42021004 and 92044302).

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

**Tables**

**Table 1.** Source region division.

| Source name | The administrative areas in each region |
|---|---|
| BJ | Beijing |
| TJ | Tianjin |
| HB | Hebei |
| SX | Shanxi |
| SD | Shandong |
| HN | Henan |
| AH | Anhui |
| JS | Jiangsu |
| NWCHN | Including Shaanxi, Gansu, Ningxia, Qinghai, Inner Mongolia and Xiangjiang |
| MONGOLIA | MONGOLIA |
| NECHN | Including Heilongjiang, Liaoning, Jilin |
| SWCHN | Including Yunnan, Guizhou, Sichuan, Chongqing and Xizang |
| SECHN | Including Jiangxi, Fujian and Taiwan |
| SCHN | Including Hunan, Hubei, Guangdong and Guangxi |
| KOREA | Including North Korea and South Korea |
| JAPAN | JAPAN |
| RUSSIA | RUSSIA |
| VIETNAM | VIETNAM |
| ZJ | Zhejiang |
| SH | Shanghai |
| OCEAN | Including Bohai Sea, Yellow Sea, East Sea, South Sea and Western Pacific |

**Table 2.** Major configuration options of WRF-Chem used for this study.

| Item | Selection | Reference |
|---|---|---|
| PBL scheme | MYJ scheme | Janjic (2002) |
| Microphysics scheme | Lin scheme | Lin et al. (1983) |
| Longwave radiation scheme | RRTM | Iacono et al. (2008) |
| Shortwave radiation scheme | RRTM | Iacono et al. (2008) |
| Land surface scheme | Noah land surface model | Chen and Dudhia (2001) |
| Dry-deposition scheme | Wesely scheme | Wesely (1989) |

**Table 3.** Statistical indicators for evaluating the simulation results.

| Variables | R | IOA | MB | RMSE | MNB | MFB | TE |
|---|---|---|---|---|---|---|---|
| T (°C) | 0.93 | 0.95 | -0.25 ([-0.5, 0.5]) | 1.88 ($\leq 2.0$) | -0.01 ([-0.15, 0.15]) | -0.002 ([-0.6, 0.6]) | 1.44 ($\leq 2.0$) |
| Ua (m·s$^{-1}$) | 0.6 | 0.75 | -0.36 ([-0.5, 0.5]) | 1.68 ($\leq 2.0$) | **-2.32** | **2.01** | 1.26 ($\leq 2.0$) |
| Va (m·s$^{-1}$) | 0.76 | 0.85 | -0.01 ([-0.5, 0.5]) | 1.82 ($\leq 2.0$) | -0.02 ([-0.15, 0.15]) | 0.009 ([-0.6, 0.6]) | 1.41 ($\leq 2.0$) |
| BC (μg·m$^{-3}$) | 0.51 | 0.70 | -0.42 | 1.47 | -0.23 | -0.06 | 1.02 |
| PM$_{2.5}$ (μg·m$^{-3}$) | 0.73 | 0.93 | -11.2 | 33.79 | -1.29 | -0.06 | 26.48 |

Values that do not meet the threshold criteria are shown in bold.



**Figures**

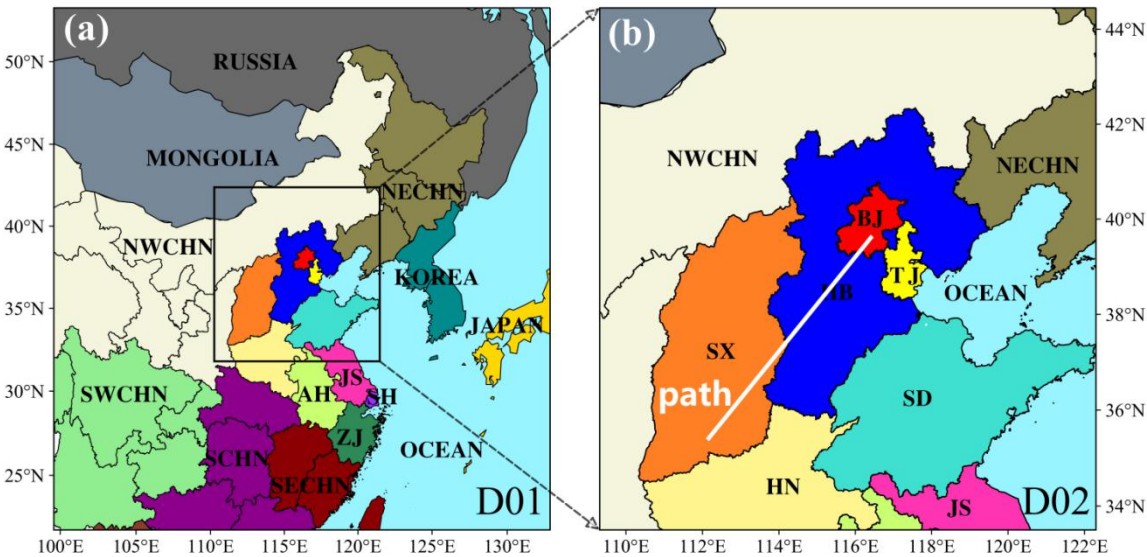

**Figure 1.** Model domain (the white line is the path of the vertical cross section in Section 3.3.2).

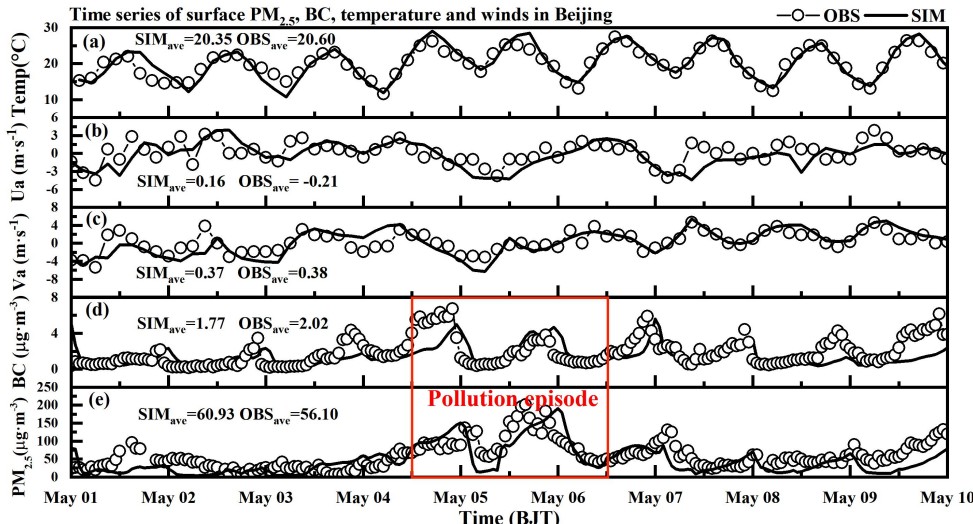

**Figure 2.** Time series of (a) temperature, (b) Ua, (c) Va, (d) BC and (e) PM$_{2.5}$ at Beijing station. The red box indicates the pollution episode in this study.

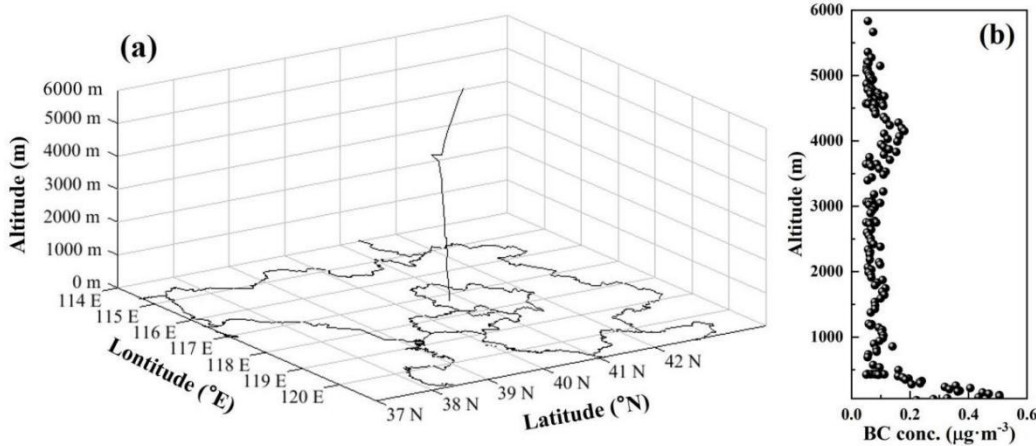

**Figure 3.** (a) Aircraft routes and (b) the BC profile observed at 10:00-11:00 BJT.

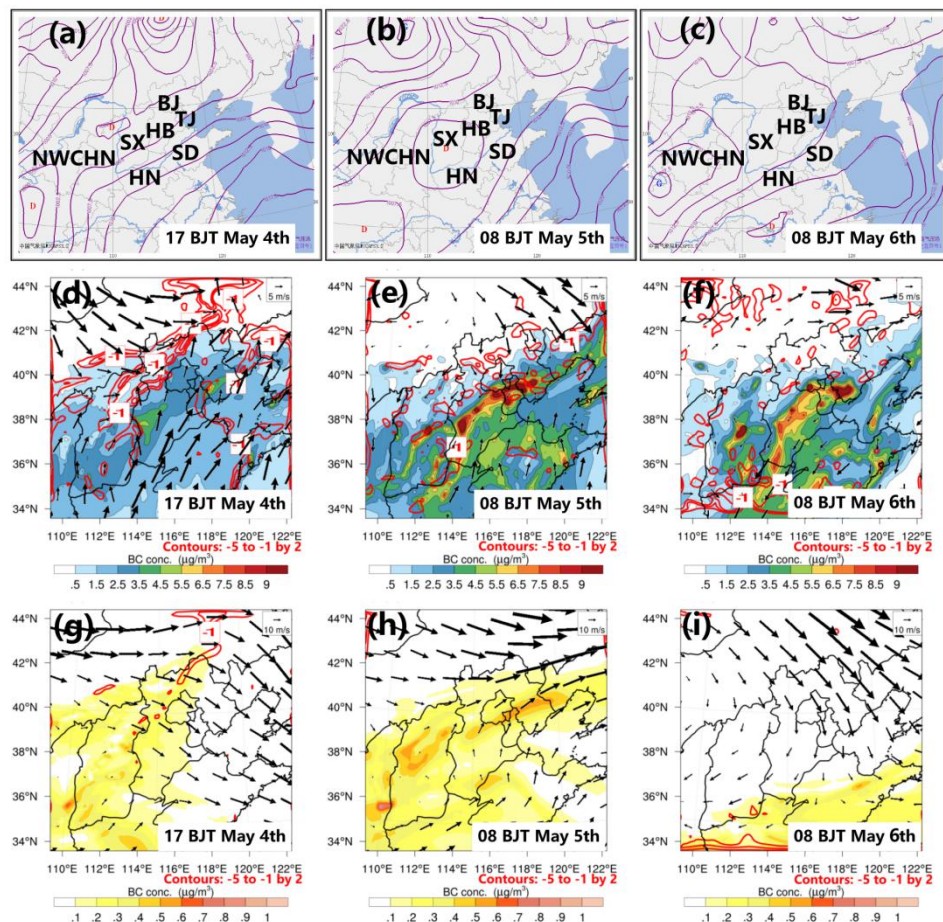

**Figure 4.** (a)-(c) Surface synoptic patterns, (d)-(f) BC concentration and wind vector at the surface, and (g)-(i) BC concentration and wind vector in the troposphere (~4000 m) at 17:00 BJT on May 4th, 08:00 BJT on May 5th, and 08:00 BJT on May 6th. The red line represents the convergence region (divergence < 0; unit: $\times 10^{-6}\,\mathrm{s^{-1}}$).

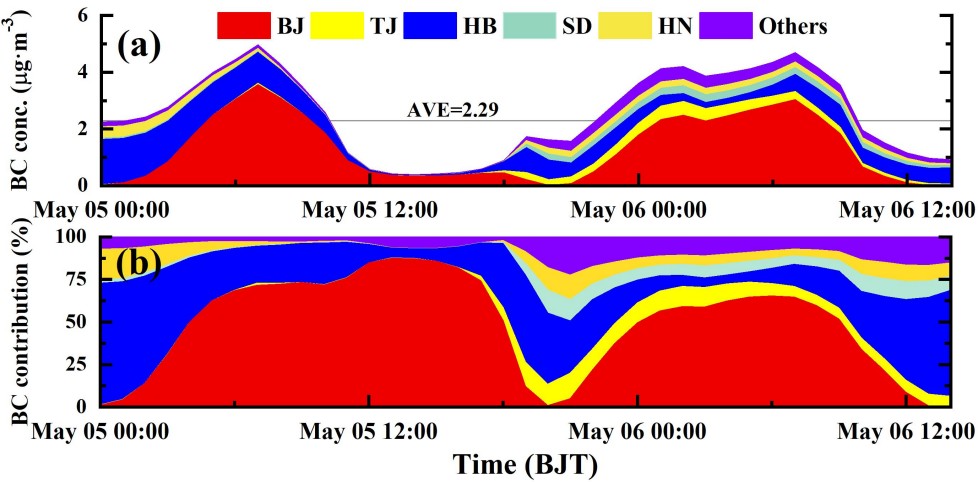

**Figure 5.** (a) BC concentration and (b) contribution rate of each source region to BC in BJ during the pollution episode. Legend is the same as that in Figure 1.

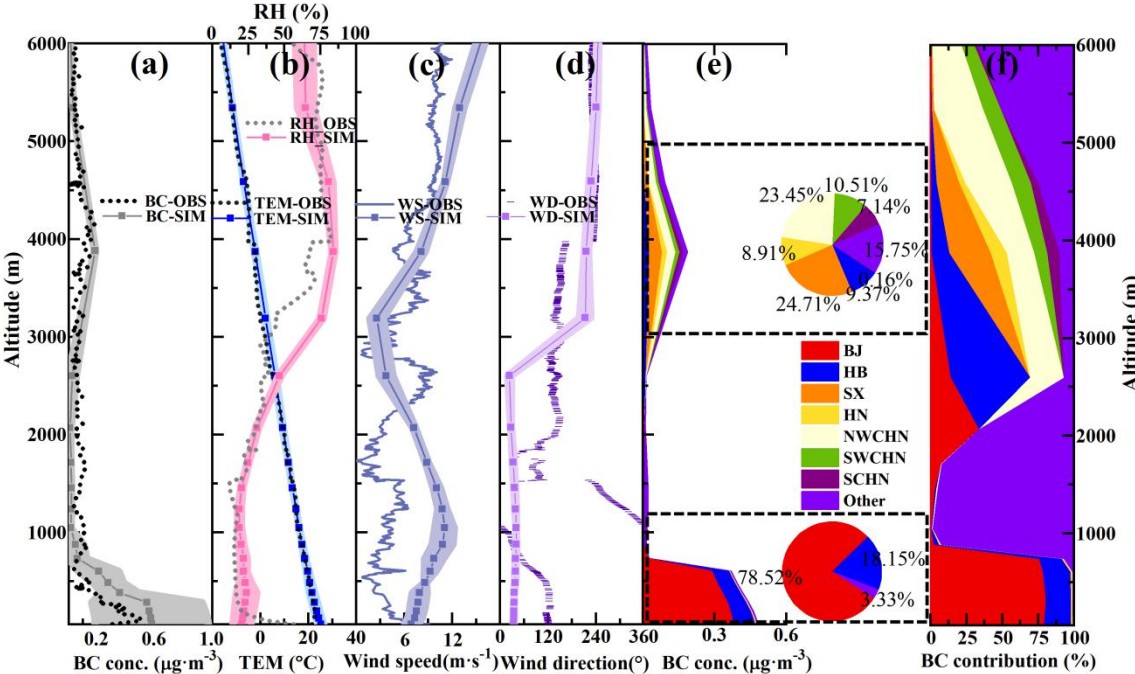

**Figure 6.** Model validation of (a) vertical BC, (b) T and RH, (c) WS, (d) WD, (e) BC concentration of each source region, and (f) contribution rate of each source region at 10:00-12:00 BJT on May 5th. The shading indicates the modelling standard deviation. Legend is the same as that in Figure 1.

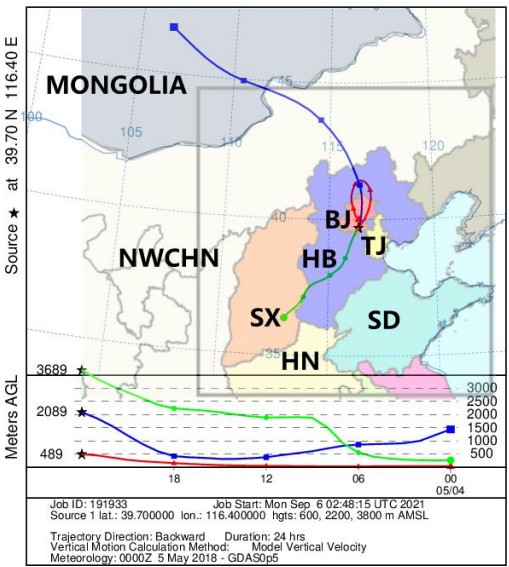

**Figure 7.** Backward trajectory (24 h) of the air mass at 08:00 BJT on May 5th.

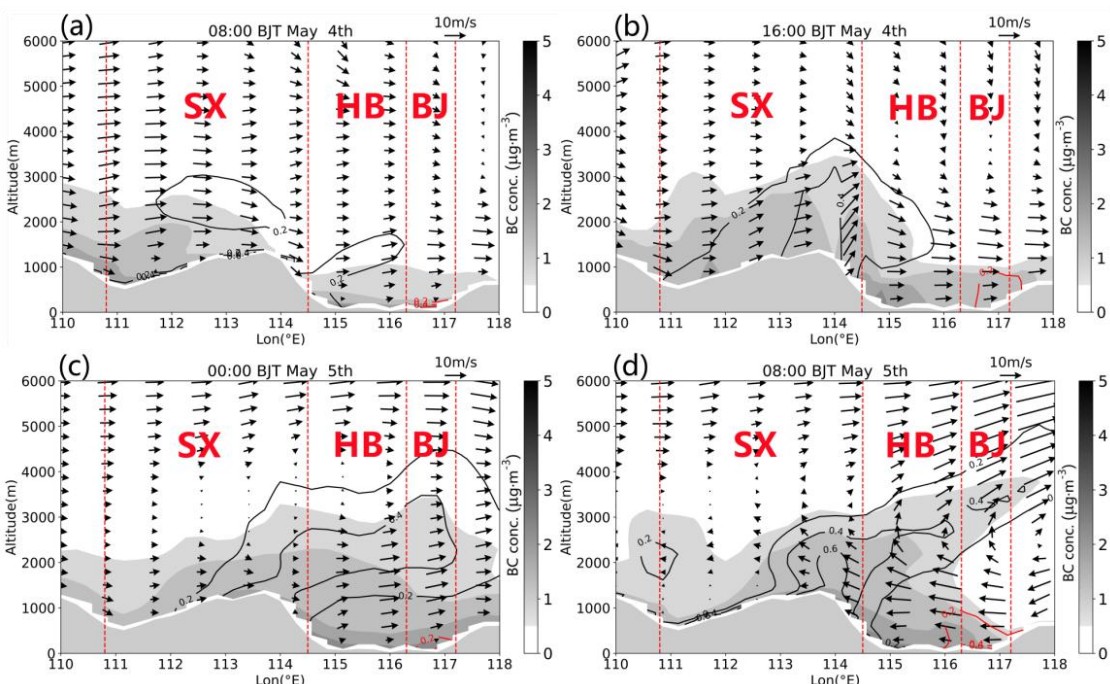

**Figure 8.** Vertical cross sections of BC concentration from SX to BJ and wind vectors (arrows) where the vertical speed is multiplied by 100 at (a) 08:00 BJT on May 4th, (b) 16:00 BJT on May 4th, (c) 00:00 BJT on May 5th, and (d) 08:00 BJT on May 5th. The black line is the BC contour lines of SX plus NWCHN, and the red line is the BC contour line of BJ.


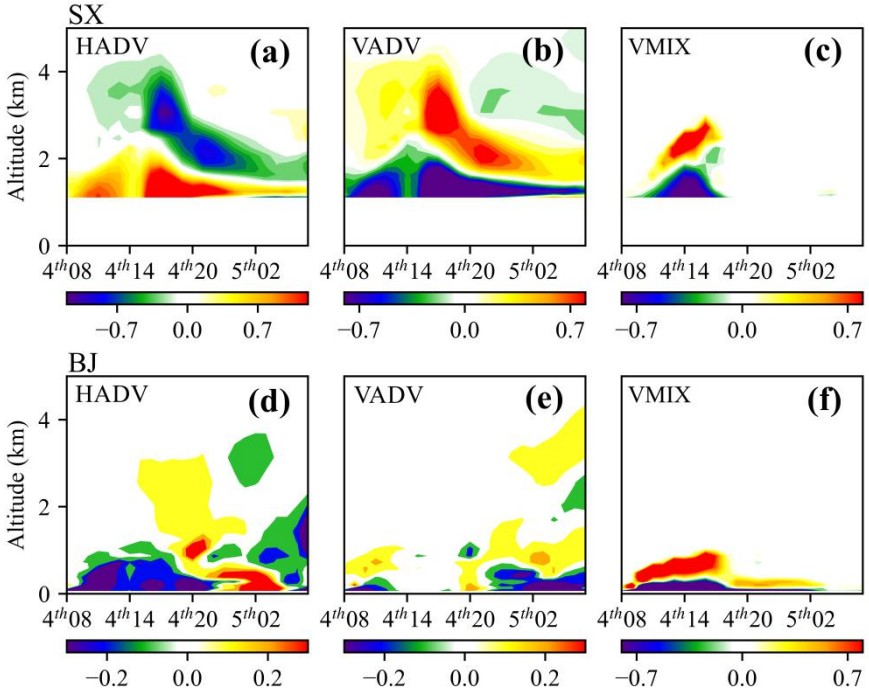

**Figure 9.** Contributions of horizontal advection (HADV), vertical advection (VADV), and vertical mixing (VMIX) processes to vertical BC concentrations in SX and BJ from 08:00 BJT on May 4th to 08:00 BJT on May 5th.


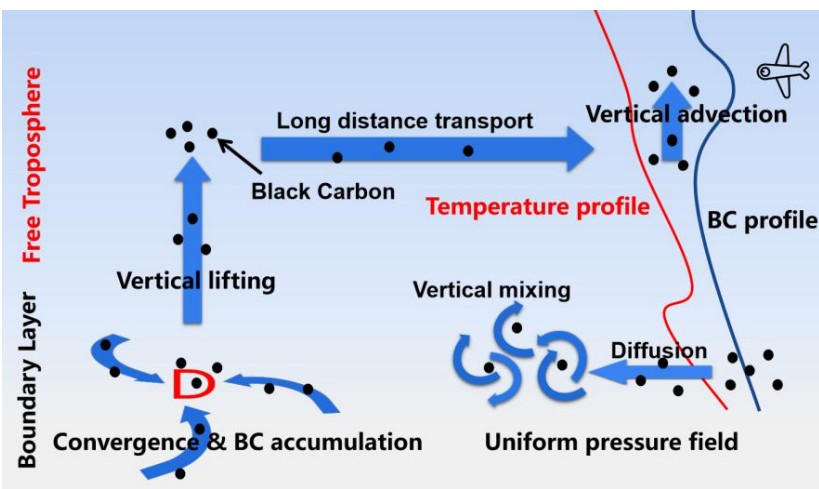

**Figure 10.** Formation mechanism of BC peak in the free troposphere.