# Peer review of "A black carbon peak in the free troposphere of Beijing induced by cyclone lifting and transport from Central China"

_Atmospheric Chemistry and Physics, 2021_

## Referee Comment (RC1)

The paper "A black carbon peak in the free troposphere of Beijing induced by cyclone lifting and transport from Central China", based on the Weather Research and Forecasting model coupled with chemistry (WRF-Chem) and focusing on BC, investigates the formation mechanism and regional sources of a BC peak in the free troposphere observed by aircraft flight in Beijing (BJ) on May 5th, 2018. The study falls within the scope of ACP. The manuscript is well-written/structured, the presentation clear, the language fluent. I would recommend publishing in ACP following major revision in specific aspects.

Comments:

Section "Model configuration" – please provide more references and extended description on the model and approach selected. In parts of the manuscript, previous studies are mentioned, without briefly providing discussion on the corresponding outputs of the studies, how do the outputs are different or beneficial or support the present study (e.g. "Previous studies have used a similar technique to study the source of air pollutants, such as PM2.5 and O3 (Gao et al., 2016; Zhang et al., 2017b; Gao et al., 2020)". Please elaborate on the introduction also.

Section "Tracking observed BC from the surface to the free troposphere" – The study implements airborne Black Carbon measurements as the main observational reference for the validation of the model, while at the same time the observed case corresponds to the study-case related to the scientific question. Therefore, the authors should provide more information on the flight and the performed measurements. Was the flight performed under a collaborative field campaign, which were the objectives, how many flights were performed, which instruments were mounted on the airplane, passive, active, or in-situ, algorithms used, and most important, how large were the uncertainties errors of the performed measurements? Please include references when and where necessary.

Section "Tracking BC sources at the surface" – More information needed on the PM2.5 measurements (instruments used - network, uncertainties/errors, density of sensors, references). Moreover, upon spatial and temporal comparison of model BC and PM2.5, in which way spatially and temporarily the presented datasets were pre-processed, in order to be able to compare SIM and OBS. Please add a comparison between SIM and OBS including discussion on the relative differences and the uncertainties, including references when and where necessary.

Section "Tracking BC sources vertically" – Regarding the validation of the model, BC model outputs and Observations seems relatively in agreement. However, more information needed. The authors should include uncertainties in the observations, according to the airborne errors of the performed measurements, discuss possible errors/uncertainties in the model outputs and provide a more in-depth analysis. For instance, the authors could provide the relative difference between SIM and OBS, for each of the presented parameter. Moreover, although similarities are evidence, so are disagreements between SIM and OBS (e.g. BC at 500m, WS below 2km, WD at 1.25km), not discussed at all in the section. Please quantify and address the comparison results, including discussion, justification, and references when and where necessary. Phrases such "model grasped the key features", should be followed by quantitative estimation of the envelope of the uncertainties (e.g. within # SD, ± error).

Section "Air mass trajectory analysis" - HYSPLIT computes the air parcel's transport and dispersion from a source region and describes where the air parcel will go. In the framework of the study, trajectories reaching below the PBL are provided. Please provide trajectories of air masses reaching at different altitudes above BJ,

corresponding to the altitudes/measurements provided by the flight. Moreover, based on CALIPSO CALIOP observation, the 5th of May 2018 is a day of extended cloud coverage over E. China. Have the authors made any analysis on possible removal of BC through wet-deposition prior the air-masses reach the flight region? The authors could implement observations on the presence of aerosol to the central-eastern China (i.e. AERONET and AE, MODIS AOD and AE over ocean/ CALIOP volume/particle depolarization ratio).

The title needs elaboration to reflect better the scientific content of the performed study, to be more accurate.

The authors should use error-bars in the figures as a metric of the uncertainty (e.g. Figures 5 and 6).

Sections "Physical process analysis" could be merged with the work performed to provide Figure 4 results, towards higher cohesion and continuity in the manuscript.

It would be beneficial for the manuscript to include a flowchart showing the instruments, datasets, methodologies and key references for the comparison followed by the authors. The entire process can be summarized there along with the methodology requirements followed e.g. the spatial - temporal constraints, screening requirements. In sections where the information exists in the manuscript, it is scattered among the sections. Furthermore, I suggest the authors to provide the collocation criteria (both spatial and temporal), since the SIM, flight obs, and PM2.5 datasets are very different.

---

## Author Response (AR1)

Dear editor and referees,

Thank you for giving us an opportunity to revise our manuscript (acp-2021-339). We appreciate your constructive comments and suggestions. We have studied these comments and suggestions carefully and made revisions to the manuscript. These comments and the corresponding replies are listed below. In the revised manuscript, we invited one new coauthor, Jinhui Gao, to help describe the BC tagging technique and process analysis.

The referees' comments are highlighted in grey. Following the comments are our responses (normal text) and revised text in the manuscript (**led by line number**). Revisions in the manuscript are coloured in red. The revised manuscript with tracked changes is attached at the end of this file.

With regards,

Zhenbin Wang, Bin Zhu*, and all coauthors.

**Replies to RC #1**

**1.** Section "Model configuration"—please provide more references and extended description on the model and approach selected. In parts of the manuscript, previous studies are mentioned, without briefly providing discussion on the corresponding outputs of the studies, how do the outputs are different or beneficial or support the present study (e.g. "Previous studies have used a similar technique to study the source of air pollutants, such as $PM_{2.5}$ and $O_3$ (Gao et al., 2016; Zhang et al., 2017b; Gao et al., 2020)". Please elaborate on the introduction also.

We are very grateful for your patient read, valuable and insightful advises for our paper. For the models and methods used in this study, we have added an extra 4 references and described the principles of the models in more detail. Moreover, in this revision, we further briefly describe the "previous studies" mentioned in the manuscript and the benefits of their findings for this study. The specific modifications are as follows:

**Lines 107–108 in the revised manuscript (Model description)**

The model used in this study, WRF-Chem 3.9.1.1, is coupled with a BC-tagging technique (Wang et al., 2014; Yang et al., 2017; Yang et al., 2018; Fang et al., 2020; Zhu et al., 2020), which is similar to.....

**Lines 111–113 in the revised manuscript (Model description)**

We divide the model domain into 20 geographic source regions before running the model, so the BC concentration of the target region is equal to the sum of the concentrations of BC from all source regions:

$$C = \sum_{i=1}^{20} C_i \qquad (1).$$

**Lines 115–121 in the revised manuscript (Model description)**

The BC concentration in each source region can be expressed as follows:

$$\Delta C_i = \Delta Chem_i + \Delta Phy_i + \Delta Emis_i \qquad (2),$$

where $\Delta Chem_i$, $\Delta Phy_i$ and $\Delta Chem_i$ represent the BC concentration produced by chemical, physical and emission processes, respectively. However, BC is not included in the calculation of chemical reactions, so the value of $\Delta Chem_i$ is 0. $\Delta Phy_i$ represents the concentration of BC changed by model physical processes, including horizontal advection, vertical advection, vertical mixing, dry deposition, wet deposition and convection.

**Lines 131–134 in the revised manuscript (Model description)**

Previous studies have used a similar technique to study the source of air pollutants, such as PM$_{2.5}$ and O$_3$, and the results all show that regional transport is an important factor of PM$_{2.5}$ and O$_3$ pollution in BJ (Gao et al., 2016; Zhang et al., 2017b; Gao et al., 2020). In addition, the air pollutants in BJ are mainly from BJ, TJ and HB, and the high concentration area presents a banded distribution feature from southwest to northeast, which is consistent with the analysis of BC in this study.

**2.** Section "Tracking observed BC from the surface to the free troposphere"— The study implements airborne Black Carbon measurements as the main observational reference for the validation of the model, while at the same time the observed case corresponds to the study-case related to the scientific question. Therefore, the authors should provide more information on the flight and the performed measurements. Was the flight performed under a collaborative field campaign, which were the objectives, how many flights were performed, which instruments were mounted on the airplane, passive, active, or in-situ, algorithms used, and most important, how large were the uncertainties errors of the performed measurements? Please include references when and where necessary.

Thank you for this valuable suggestion. We agree with your opinion that it is necessary to provide more information on the flight and the performed measurements. The airborne measurements were conducted mainly with the support of the National Key Research and Development Program of China (The climate and atmospheric environment impact of black carbon from residential and agricultural sources). One of the main objectives of the program was concentrated on the vertical profiles of black carbon and its sources. We added more details about aircraft flight times, tracks, speeds and instruments carried by the aircraft to the manuscript, along with corresponding references. The specific modifications are shown below:

**Lines 97–105 in the revised manuscript (Section "Aircraft observation platform")**

In May 2018, we carried out a total of 7 aircraft observations in BJ and surrounding areas (Figure S1), with each observation time between 10:00 and 12:00 (Beijing Time, BJT). The airborne measurements in this study were performed on a King Air 350 aircraft observation platform with a true speed of approximately 250-300 km/h. An Aircraft Integrated Meteorological Measurement System (AIMMS-20, Aventech, Canada) was used to measure meteorological parameters in situ, including temperature (T), relative humidity (RH), wind speed (WS) and wind direction (WD), with a time resolution of 1s. An SP2 was also installed on the aircraft observation platform to measure

the vertical distribution of BC. The operation of most flights was carried out to avoid clouds where possible, which could effectively avoid the wet deposition of BC. More detailed information about the aircraft platform can be found in Tian et al. (2019), Hu et al. (2020) and Zhao et al. (2019&2020).

[Figure]

**Figure S1.** Flights tracks in BJ, May 2018.

**Lines 172–173 in the revised manuscript (Tracking BC from the surface to the free troposphere)**

During the aircraft observation on May 5th, 2018, we found high BC values both near the ground and in the free troposphere (Fig. 3).

**3.** Section "Tracking BC sources at the surface"—More information needed on the PM$_{2.5}$ measurements (instruments used - network, uncertainties/errors, density of sensors, references). Moreover, upon spatial and temporal comparison of model BC and PM$_{2.5}$, in which way spatially and temporarily the presented datasets were pre-processed, in order to be able to compare SIM and OBS. Please add a comparison between SIM and OBS including discussion on the relative differences and the uncertainties, including references when and where necessary.

Thank you for your valuable suggestion. We added information about observation instruments and acquisition methods of PM$_{2.5}$ data on the ground and provided relevant references in the "Ground-based observation data" section. The time series of PM$_{2.5}$ and BC concentrations are shown in Fig. 2, and their trends are quite consistent. Following your advice, we compared the spatial distribution characteristics of model BC and PM$_{2.5}$ (Fig. S4) and found that they have similar spatial distribution characteristics, both near the surface and in the free troposphere. In addition, we performed a further analysis of the discrepancy between SIM and OBS in the "Model validation" section. The specific modifications are shown below:

**Lines 84–95 in the revised manuscript (Section "Ground-based observation data")**

PM$_{2.5}$ data were obtained from the China air quality online monitoring and analysis platform (https://www.aqistudy.cn/) provided by the Ministry of Ecology and Environment of China (http://106.37.208.233:20035/), which includes 1500 monitoring stations. Each site was equipped with tapered element oscillating microbalance instruments (TEOM, RP1400 Model, Thermo Scientific Company, USA) using the micro-oscillating balance method and the β-absorption method to measure PM$_{2.5}$ concentration at a resolution of 0.1 μg·m$^{-3}$ (Zhang and Cao, 2015). More details about the measurement methods, accuracy, and uncertainties are described in China Environmental Protection Standards HJ 653-2013 (http://www.cnemc.cn/jcgf/dqhj/201711/W020181008687887167307.pdf). Many studies on air pollution, particularly PM$_{2.5}$ pollution, have obtained their PM$_{2.5}$ data from this platform, such as Kang et al. (2021) and Hou et al. (2019, 2020). BC data were collected by the single particle soot photometer (SP2, Droplet Measurement Technologies, Inc., USA) that was equipped with an external inlet, the Model 1200 passive isokinetic inlet (Brechtel Manufacturing Inc., USA), which could deliver a sample flow of 150 lpm with an air speed of 100 m·s$^{-1}$ with a sample efficiency of 95% for particle sizes between 0.01 and 6 μm.

**Lines 160–162 in the revised manuscript (Model validation)**

The MB of T was -0.25, which suggested that the mean bias of T was within 1 ℃, and the MB of Ua and Va were -0.36 and -0.01, respectively, suggesting that the model results deviated from the observations to a small extent.

**Line 165 in the revised manuscript (Model validation)**

The MNB and MFB of the other meteorological factors were within the scope of their benchmark.

**Lines 167–169 in the revised manuscript (Model validation)**

The MB values of BC and $PM_{2.5}$ were -0.42 and -11.2, respectively, which demonstrated that the deviation in MB between BC models and observations was less than 1 $\mu g \cdot m^{-3}$, and the deviation in $PM_{2.5}$ was approximately 11 $\mu g \cdot m^{-3}$.

**Lines 176–177 in the revised manuscript (Tracking BC sources at the surface)**

...both the spatial and temporal distributions (near-surface and in the free troposphere) of BC and $PM_{2.5}$ exhibited highly similar characteristics (Fig. S4).

[Figure]

**Figure S4.** Near-surface BC (a, e, i) and PM$_{2.5}$ (b, f, j). BC (c, g, k) and PM$_{2.5}$ (d, h, k) in the free troposphere.

**4.** Section "Tracking BC sources vertically"—Regarding the validation of the model, BC model outputs and Observations seems relatively in agreement. However, more information needed. The authors should include uncertainties in the observations, according to the airborne errors of the performed measurements, discuss possible errors/uncertainties in the model outputs and provide a more in-depth analysis. For instance, the authors could provide the relative difference between SIM and OBS, for each of the presented parameter. Moreover, although similarities are evidence, so are disagreements between SIM and OBS (e.g. BC at 500m, WS below 2km, WD at 1.25km), not discussed at all in the section. Please quantify and address the comparison results, including discussion, justification, and references when and where necessary. Phrases such "model grasped the key features", should be followed by quantitative estimation of the envelope of the uncertainties (e.g. within # SD, ± error).

Thank you for your patient read and insightful comment. For the SIM and OBS in the vertical direction, the frequency of the observation instruments carried by the aircraft is 1 s, so there are more than 570 observations from the surface to the free troposphere, but only nearly 20 model outputs correspond to the observation altitude. Therefore, owing to the difference in the amount of data, it was difficult to quantitatively calculate the statistical characteristics of the elements in the vertical direction, and we mainly qualitatively compared the trends and numerical magnitudes of the SIM and OBS. A more in-depth discussion on the relative difference between SIM and OBS is given in the third response. In addition, we explained the disagreements between SIM and OBS that you mentioned and modified the phrases such as "model grasped the key features". The specific modifications are shown below:

**Lines 200–205 in the revised manuscript (Tracking BC sources vertically)**

The vertical distributions of BC and meteorological factors were well simulated, especially BC, T and RH (Fig. 6a and b). However, the WS under 2 km and WD at 1.25 km did not well show agreements between SIM and OBS, which could be attributed to local circulations such as land-sea and valley breezes across the complex terrain (Igel et al., 2018; Quan et al., 2020; Zhang et al., 2021). In this study, we focus more on the effect of upper-level winds on the transport of BC, and the winds at the altitude of 4000 m are significantly better modelled than the winds within the boundary layer. Overall, the model results are also acceptable in the vertical direction.

**5.** Section "Air mass trajectory analysis" — HYSPLIT computes the air parcel's transport and dispersion from a source region and describes where the air parcel will go. In the framework of the study, trajectories reaching below the PBL are provided. Please provide trajectories of air masses reaching at different altitudes above BJ, corresponding to the altitudes/measurements provided by the flight. Moreover, based on CALIPSO CALIOP observation, the 5th of May 2018 is a day of extended cloud coverage over E. China. Have the authors made any analysis on possible removal of BC through wet-deposition prior the air-masses reach the flight region? The authors could implement observations on the presence of aerosol to the central-eastern China (i.e. AERONET and AE, MODIS AOD and AE over ocean/ CALIOP volume/particle depolarization ratio).

Thank you for this valuable suggestion. We reused the HYSPLIT model for backward trajectory analysis of air masses reaching altitudes of 600 m, **2200 m (new)** and 3800 mover Beijing (with intervals of 1600 m, Fig. 7), and the results are consistent with the previous results and explain the origin of air masses in the middle atmosphere. **In addition, the operation of most flights was carried out to avoid clouds where possible, which could effectively avoid the wet deposition of BC (Lines 103–104).** Moreover, historical weather indicates that no precipitation was generated at the time of observation, and the subcloud clearing of BC is weak. Therefore, the possible removal of BC through wet deposition is negligible. The specific modifications are shown below:

**Lines 217–218 in the revised manuscript (Section "Air mass trajectory analysis")**

... to analyse the backward trajectory (24 h) of air masses reaching the altitudes of 600, 2200 and 3800 m above BJ, with intervals of 1600 m, at 08:00 BJT on May 5th (Fig. 7).

**Lines 221–223 in the revised manuscript (Section "Air mass trajectory analysis")**

In addition, the air mass reaching at the altitude of approximately 2000 m above BJ (blue line) was from Mongolia, a source region outside of D02, so the BC source tracing results in a predominantly Other source region (Fig. 6e, the Other source region includes Mongolia and Inner Mongolia).

[Figure]

Figure 7. Backward trajectory (24 h) of the air mass at 08:00 BJT on May 5th.

**6.** The title needs elaboration to reflect better the scientific content of the performed study, to be more accurate.

We agree with your suggestion. We changed the title to "A black carbon peak and its sources in the free troposphere of Beijing induced by cyclone lifting and transport from Central China" to reflect the scientific objectives of our research, i.e., the vertical profiles of black carbon and its sources.

**7.** The authors should use error-bars in the figures as a metric of the uncertainty (e.g. Figures 5 and 6).

Thank you for this valuable suggestion. We have used error bars as a metric of the modelling uncertainty in the new Fig. 6. The aircraft observations were carried out between 10:00 and 12:00, and the model results were set to output hourly. Therefore, the modelling results at 10:00, 11:00 and 12:00 are selected to show the range of BC concentrations around the period of the aircraft observation, as shown in the new Figure 6. However, this process was observed only once by the aircraft, so error bars could not be used for observation profiles. Regarding Figure 5, in this case, it was a time series of BC source tracing, so error bar analysis cannot be used.

[Figure]

Figure 6. Model validation of (a) vertical BC, (b) T and RH, (c) WS, (d) WD, (e) BC concentration of each source region, and (f) contribution rate of each source region at 10:00–12:00 BJT on May 5th. The shading indicates the modelling standard deviation. Legend is the same as that in Figure 1.

**8.** Sections "Physical process analysis" could be merged with the work performed to provide Figure 4 results, towards higher cohesion and continuity in the manuscript.

Thank you for this valuable suggestion. After careful consideration, we have decided to retain the "process analysis" section for the reasons described below. Figure 4 mainly reflects the influence of the surface synoptic situation, divergence field and wind field at different altitudes on the BC distribution during this episode. However, the "Physical process analysis" section mainly describes the dominant transport processes of BC, which can be combined with the analysis in Fig. 4. The specific modifications are shown below:

**Line 244–245 in the revised manuscript (Physical process analysis)**

The backward trajectory shows that the air mass in the free troposphere originated from SX, while that within the boundary layer was from BJ; therefore, we further quantified the dominant transport processes of BC...

**Line 250–251 in the revised manuscript (Physical process analysis)**

... suggesting that convergent and upward motion existed therein (Fig. 9a and b), which also corroborates the cyclone system in the surface synoptic patterns and near-surface convergence areas (Fig. 4a and d).

**Line 256 in the revised manuscript (Physical process analysis)**

... showing that BC was transported from SX by westerly winds, which is also consistent with the analysis of BC transport in Figure 4h.

**9.** It would be beneficial for the manuscript to include a flowchart showing the instruments, datasets, methodologies and key references for the comparison followed by the authors. The entire process can be summarized there along with the methodology requirements followed e.g. the spatial - temporal constraints, screening requirements. In sections where the information exists in the manuscript, it is scattered among the sections. Furthermore, I suggest the authors to provide the collocation criteria (both spatial and temporal), since the SIM, flight obs, and PM$_{2.5}$ datasets are very different.

Thank you for this valuable suggestion. We have added a technical flow chart to the manuscript that describes the observational data, including ground-based observations, aircraft observation platforms, instruments used, model versions, model-driven data and parameterized scheme settings.

[Figure]

**Figure S7.** Technical flowchart.

The model validation in North China, particularly in HN and SX (source regions), is presented in Figures S2 and S3 and Tables S1 and S2 in the supplement, and the modelling profiles of BC and meteorological factors are also presented in Fig. 6 in Section 3.2.2.

[Figure]

**Figure S2**. Model validation in HN.

**Table S1.** Statistical indicators for evaluating the simulation results in HN.

| Variables | IOA | MB | RMSE | MNB | MFB | TE |
|---|---|---|---|---|---|---|
| T (°C) | 0.98 | -0.24 ([-0.5, 0.5]) | 2 (≤ 2.0) | -0.04 ([-0.15, 0.15]) | -0.03 ([-0.6, 0.6]) | 1.58 (≤ 2.0) |
| RH | 0.99 | -8.05 | **11.62** | -0.15 ([-0.15, 0.15]) | -0.11 ([-0.6, 0.6]) | 9.43 |
| WS (m·s⁻¹) | 0.55 | 1.45 | **2.17** | **0.57** | 0.38 ([-0.6, 0.6]) | 1.73 |
| WD (m·s⁻¹) | 0.92 | -14.8 | 93.54 | -0.1 ([-0.15, 0.15]) | -0.4 ([-0.6, 0.6]) | **54.11** |
| BC (μg·m⁻³) | 0.64 | 0.52 | 2.2 | 0.24 | 0.18 | 1.49 |
| PM₂.₅ (μg·m⁻³) | 0.71 | 9.72 | 33.84 | 0.2 | 0.15 | 26.26 |

Values that do not meet the threshold criteria are shown in bold.

[Figure]

**Figure S3**. Model validation in SX.

**Table S2.** Statistical indicators for evaluating the simulation results in SX.

| Variables | IOA | MB | RMSE | MNB | MFB | TE |
|---|---|---|---|---|---|---|
| T (°C) | 0.99 | -0.25 ([-0.5, 0.5]) | 1.89 (≤ 2.0) | -0.05 ([-0.15, 0.15]) | 0.01 ([-0.6, 0.6]) | 2.55 (≤ 2.0) |
| RH | 0.97 | -4.26 | **14.54** | -0.09 ([-0.15, 0.15]) | -0.01 ([-0.6, 0.6]) | 11.52 |
| WS (m·s⁻¹) | 0.54 | 1.44 | **2.43** | **0.52** | 0.37 ([-0.6, 0.6]) | 1.83 (≤ 2.0) |
| WD (m·s⁻¹) | 0.75 | 16.09 | 140.23 | 0.08 ([-0.15, 0.15]) | 0.2 ([-0.6, 0.6]) | **98.91** |
| BC ($^{\mu}$g·m⁻³) | 0.77 | 0.17 | 0.46 | 0.22 | 0.16 | 0.28 |
| PM₂.₅ ($^{\mu}$g·m⁻³) | 0.69 | 3.66 | 32.29 | 0.08 | 0.24 | 25.82 |

Values that do not meet the threshold criteria are shown in bold.

**2.** Usually, high BC is related to impact of biomass burning because these emissions are easilly lifted to high altitudes in high temperature without atmospheric dynamic uplifting. I suggested that authores investigate the impact of biomass burning in this study. At least, fire counts from satellites.

Thank you for this valuable suggestion. Following your advice, we have used MODIS satellite products to research the distribution of fire points during the study period (4th–6th May). The MODIS Fire and Thermal Anomalies product is available from the combined Terra and Aqua (MCD14) satellite product. The spatial and temporal resolutions are 1 km daily. The thermal anomalies are represented as red points. **We found that the fire points were mainly concentrated in Jiangsu and Liaoning Provinces on May 4th and mainly in Liaoning Province on May 6th, with almost no fire points on May 5th. There were few fire points in and around Beijing during these three days, especially on the air mass pathway to Beijing (Fig. 8 in the manuscript). In addition, the busy agricultural season in North China is generally from late May to early June (Deng et al., 2006).** Therefore, we concluded that biomass burning made a minor contribution to the BC over Beijing in this study period.

[Figure]

**Figure S5**. Fire maps on (a) May 4th, (b) May 5th and May 6th.

**3.** Section 3.2.1 more dynamic conditions are suggested to conduct. For example, the convergence of BC/PM2.5 at different altitudes in each phase of this episode. It's better to provide a comprehensive dynamic driving forces of regional transport.

Thank you for this valuable suggestion. The divergence field is indeed more helpful to reflect the dynamic driving forces of BC lifting and regional transport, so we have added divergence fields in the near surface (Fig. 4e, f and g) and the free troposphere (Fig. 4h, i and j). **Figure 4 shows that the near-surface convergence area was mainly in SX from May 4th to 5th, corresponding to the position where the cyclone appeared, indicating that convergent uplift of surface BC existed in SX, which was consistent with the cross-sectional analysis in Section 3.3.2. The convergence area is not obvious in the free troposphere, and BC is transported by westerly winds.** In addition, the integral results from the surface to the free troposphere are shown in Figure S6 of the supplement, and the main convergence regions are also over SX and HN. The specific modifications are shown below:

**Lines 180–183 in the revised manuscript (Tracking BC sources at the surface)**

At the same time, it can be seen that the near-surface convergence areas were mainly in SX from May 4th to 5th (Fig. 4d and e), corresponding to the position where the cyclone appeared, indicating that convergence uplift of surface BC existed in SX, which was consistent with the cross-section analysis in Section 3.3.2.

**Line 185 in the revised manuscript (Tracking BC sources at the surface)**

Moreover, there was almost no convergence but an obvious 'transport channel' from SX to BJ...

[Figure]

Figure 4. (a)-(c) Surface synoptic patterns, (d)-(f) BC concentration and wind vector at the surface, and (g)-(i) BC concentration and wind vector in the troposphere (~4000 m) at 17:00 BJT on May 4th, 08:00 BJT on May 5th, and 08:00 BJT on May 6th. The red line represents the convergence region (divergence < 0; unit: $\times 10^{-6}\,s^{-1}$).

**4.** 3.3.1 what is the input data of HYSPLIT?

Thank you for your valuable suggestion. The website of the HYSPLIT model is presented in parentheses (https://ready.arl.noaa.gov/hypub-bin/trajtype.pl?runtype=archive). We use the NCEP Global Data Assimilation System (GDAS) half-degree archive (0.5 degree, global, Sep, 2007 - June, 2019) to compute the 24 h backward trajectory, and the GDAS data can be obtained from the website links in parentheses (*ftp://ftp.arl.noaa.gov/pub/archives/gdas0p5*).

**5.** LIne 169 China central seems to be cenrtal China.

Thank you for your reminder. We have corrected this typo.

**Line 225 in the revised manuscript (In this version, the original line 169 is now line 225)**

During this time, the conditions in Central China (including SX, HB and HN)...

**References**

Deng, X. P., Shan, L., Zhang, H. and Turner, N. C.: Improving agricultural water use efficiency in arid and semiarid areas of China, Agricul. Water Manag., 80, 23-40, *https://doi.org/10.1016/j.agwat.2005.07.021*, 2006.